# Gender, Age, Race and Lactose Intolerance: Is There Evidence to Support a Differential Symptom Response? A Scoping Review

**DOI:** 10.3390/nu10121956

**Published:** 2018-12-11

**Authors:** Rebecca A Lapides, Dennis A Savaiano

**Affiliations:** Nutrition Science, Purdue University, West Lafayette, IN 47906, USA; rlapides@purdue.edu

**Keywords:** Lactose intolerance, age, gender, race

## Abstract

Research evaluating the relationship between lactose intolerance (LI) symptoms and age, gender and race is reviewed. An exhaustive search was conducted on the Google Scholar and PubMed databases. The evidence suggests that women, the elderly or specific racial groups are not more susceptible to LI, but rather dose, body size and genetic differences in lactase non-persistence (LNP) are the primary drivers of intolerance symptoms.

## 1. Introduction

This scoping review summarizes the relationship between lactose intolerance (LI) symptoms and age, gender and race. Several authors claim that women, the elderly or specific racial groups suffer more from lactose intolerance beyond the expected incidence based on the genetic determinant of lactase non-persistence (LNP) [1,2,3,4,5]. It has also been demonstrated that women, the elderly or specific racial groups perceive LI at a high rate, and avoid dairy products due to this perception [1,6,7,8]. For example, Johnson et al. [1] suggested that 1/3 of African Americans perceive themselves to be LI, and Nicklas et al. [7] demonstrated that in a large sample of 3452 adults, 76.13% of the respondents who perceived themselves to be LI were female and consumed considerably lower amounts of calcium. The subjects who perceived themselves to be LI also had higher rates of chronic health conditions. Matlik et al. [8] demonstrated that young girls who perceived LI (not those who were actually LNP) had reduced calcium intakes and lower bone densities. In addition, Baadkar et al. [5] suggested that females and the elderly are more LI than males and younger populations, however, this conclusion was made after a common dose of lactose was fed, regardless of body size.

Perceptions about age, gender and race related to LI are a barrier to effective education and dietary selection. We found very limited research to support the claims of unique gender, age or race differences in LI beyond LNP differences. Rather, dose, diet (including dietary adaptation of the microbiome), body size and LNP are the likely primary drivers of intolerance symptoms. The 2010 NIH Consensus Conference on LI cites a critical need to educate LNP consumers on dietary habits that will encourage nutrient density, including consuming adequate amounts of calcium and Vitamin D. Women, the elderly and specific racial groups have particularly low calcium intakes. Encouraging appropriate dietary choices is especially critical in these populations.

We searched for original clinical research that measured symptoms of LI while also reporting demographics of the population under study. We used the following key search phrases: “lactose intolerance,” “milk intolerance,” “lactose intolerance demographics,” “milk intolerance demographics,” “lactose maldigestion,” “lactose maldigestion symptoms” and “symptoms of lactose intolerance.” Google Scholar and PubMed databases were searched through October of 2018. Included studies were published in; American Journal of Clinical Nutrition, Microbial Ecology in Health and Disease, Digestive Diseases and Sciences, the Journal of Nutrition, the Journal of the American College of Nutrition, the Indian Journal of Medical Research, the British Journal of Nutrition, Alimentary Pharmacology & Therapeutics, the Nutrition Journal, and Annals of Human Biology. Only 14 studies were identified that included symptom data that was relatable to gender, age or race demographics. The relevant studies included were published from 1974 to 2014. There were no papers beyond 2014 that satisfied the criteria. The remainder of this review summarizes those studies in chronological order, reviewing them for confounding factors including dose of lactose, dose of lactose per kg body wt., meal-feeding, lack of blinding, etc. Several studies that claimed gender differences in LI were confounded by dosing identical amounts of lactose to individuals regardless of body size. Table 1 summarizes the confounding factors from each study. The findings suggest little or no evidence to support the claim that women, the elderly or specific racial groups are more susceptible to LI, beyond the expected genetic differences in LNP.

## 2. Clinical Trials

In 1974, Stephenson and Latham reported the LI symptoms experienced by 35 healthy adult volunteers, 19 females and 16 males, 25 Caucasians and 10 non-Caucasians (as described by the authors), median age of 25 years [9]. Each subject was given a dose of 50 g of lactose dissolved in 200 mL of water in the morning after an overnight fast, and the symptoms were assessed for the next 8 h. Symptoms included diarrhea, gas, bloating, and cramps, rated as mild, moderate, or severe. LNP was present in 16/35 subjects based on a rise in blood sugar over a fasting level of less than 20 mg/100 mL. Of the non-Caucasians, eight out of ten were LNP. Fifteen out of sixteen LNP subjects reported experiencing symptoms, however, the symptom severity varied considerably. Diarrhea and bloating were common. LNP subjects reported moderate or severe symptoms more frequently than digesters. Subsequently, varying quantities of lactose and milk were ingested by the 19 digesters and 16 LNP adults on 10 separate occasions. As expected, the symptom incidence and severity were significantly higher in the LNP individuals. Nonetheless, it was determined that the majority of these LNP adults could consume approximately 15–30 g of lactose without experiencing severe symptoms. Negligible symptoms were experienced when all subjects consumed 15 g of lactose, and 66% could consume at least 30 g, suggesting that the healthy LNP adults can consume approximately 240 mL of milk on an empty stomach without experiencing significant symptoms. Unfortunately, the authors did not stratify the symptom-response data by age, gender or race. Thus, these findings, while indicating that LNP is the primary factor in symptom-response, do not support differential LI among gender, age or racial groups.

In 1979, Rorick and Scrimshaw [10] reported milk intolerance in an elderly population of 87 individuals, 20 males and 67 females, ranging in age from 60 to 97 years, with varying ethnic backgrounds [10]. Information on ethnic background, sex, age, milk attitudes, milk consumption habits, milk intolerance history and relevant medical conditions was collected. On test days, each subject omitted dairy from the breakfast meal and consumed 240 mL of a chocolate-flavored dairy drink, containing either 4.5% or 0% lactose at lunchtime with a lactose-free meal. Respiratory hydrogen expiration was used to assess LNP. Twenty-three of eighty-seven subjects were classified as LNP. The prevalence of LNP was higher in Black, Jewish and Italian subjects and the lower in subjects of northern and western European descent. A total of 54 symptoms were reported by 25 subjects and 39 of these symptoms were judged as mild, 11 as moderate and 4 as severe. Gas and bloating were most common. This early study was among the first to relate LNP to race, and provides useful background to the modern understanding of genetic differences among humans in the maintenance or loss of intestinal lactase. The authors did not provide correlative comparisons between LNP and symptom-response in this sample. While this study does not present symptom prevalence directly corresponding to the specific ethnic backgrounds, the results do suggest that 11–12 g of lactose in a glass of milk would not lead to symptoms even in elderly individuals who are lactose malabsorbers. This conclusion is made based on an ethnically diverse sample, which indicates that most individuals from various ethnic backgrounds will be able to tolerate this quantity of lactose. Thus, there is no evidence from this study to support the hypothesis that certain demographics populations experience greater symptoms after lactose ingestion. 

In 1980, Haverberg, Kwon and Scrimshaw reported lactose malabsorption and clinical symptoms in 110 adolescents, ages 14 to 19 years, of differing ethnic backgrounds [11]. Fifty-eight subjects were black, 44 were white and 8 were of Latin-American descent. Fifty grams of lactose in 250 mL of water was administered to each subject following an overnight fast to determine LNP. Subsequently, subjects were fed 240 mL and 480 mL of lactose-containing and lactose-free chocolate milk. The presence and severity of symptoms including diarrhea, abdominal pain, flatulence, and bloating were assessed. Eighty-three percent of the black subjects, 62% of Hispanic subjects and 32% of white subjects were LNP. Unfortunately, similar to the previous study, symptom data is not correlated to ethnic background. Interestingly, symptoms occurred with both lactose-containing and lactose-free milks at both doses, inconsistent with intolerance due to lactose. This not only suggests that the risk of significant symptoms after consuming 240 mL is relatively small for adolescents, but also that the symptoms experienced may be due to factors other than lactose. Therefore, the data from this study does not support that symptoms are particularly associated with adolescents or any race included in the sample. 

A continuation of this study was reported using an improved double-blind test procedure (the authors note; explanations with less emphasis on symptoms; simplified questionnaires; and a positive symptom response with only one mild or stronger symptom) with 87 subjects ranging in age from 14 to 19 years [12]. Twenty-six subjects were black, 56 were white and 5 were Asian-American. The protocol was similar to the original study [11]. Eighty-one percent of black subjects, 80% of Asian-American subjects and 36% of white subjects were LNP. Twenty-two percent of subjects reported symptoms after the 240 mL lactose-free drink, and 16% reported symptoms after the 480 mL lactose-free drink. Sixty-four percent of lactose LNP did not report any symptoms from either lactose-containing or lactose-free milks. Thirty-one percent of lactose digesters reported symptoms on at least one day of the study. The authors concluded that all adolescents could tolerate 240 mL of milk containing lactose, however, some may experience symptoms after a 480 mL dose of lactose if they have low lactase activity (LNP), and the mild symptoms experienced in this population are likely the result of other factors and not lactose malabsorption. Similar to the initial study, no evidence of a correlation between symptoms and a specific demographic group was reported. 

In 1987, Rosado, Allen and Solomons reported gastrointestinal symptoms experienced in 25 adults after consuming 360 mL of whole milk [13]. The population included 20 females and 5 males, ages 19 to 48 years, mean of 28 years, with a history of LI based on self-reported experience with symptoms of gas, abdominal cramping, or diarrhea after milk consumption. Fifteen of the subjects were Northwestern European or Scandinavian whites; nine were whites of other ancestry and two were blacks. A control group of 13 white subjects with no history of LI participated only in the questionnaire phase of the study. Subjects were fed 360 mL of whole milk containing 18 g of lactose following an overnight fast. Breath hydrogen excretion was measured at 30 min intervals for 5 h after the milk ingestion. Sixty-four percent of subjects were digesters and 36% of subjects were LNP. However, the symptom response to the milk test was the same among lactose digesters and LNPs. The authors suggest that lactose content may not determine whether symptoms are experienced after milk consumption, but rather that there may be other factors present in milk that account for symptoms. Given this claim, there is certainly no basis for any association between symptoms caused by lactose in milk and specific demographics. 

In 1993, Ito and Kimura reported the effect of lactose on fecal microflora and intolerance symptoms in 24 healthy Japanese males between the ages of 25 and 45 years [14]. Prior to the study, none of the subjects had consumed lactose-containing food or antimicrobial drugs for 2 weeks. Each subject was given a dose of 15 g of lactose in 115 mL of water once per day for 6 days. A hydrogen breath test was conducted on day 7 to assess whether or not lactose maldigestion had occurred. Twenty-two of twenty-four subjects (92%) were LNP. However, 17 of the LNP subjects did not report discomfort or diarrhea, so they were considered lactose tolerant. Thus, this study supports that Japanese men, and perhaps Japanese individuals in general exhibit more lactose maldigestion than other ethnicities, but many of these individuals may not experience symptoms, making them not LI and allowing for consumption of milk and milk products. However, the sample size was small and did not include both genders, which is a limitation. Despite maldigestion in this population, 17 of 24 subjects reported on discomfort.

Also in 1993, Johnson et al. reported on lactose tolerance in 164 African Americans ranging in age from 12 to 40 years [1]. Inclusion criteria included subject claims of intolerance to 240 mL or less dose of milk. After a 10–12 h fast, subjects were fed 25 g of lactose in 200–300 mL of water. Breath hydrogen was measured at 30 min time intervals over 3 h after ingestion and symptoms including flatulence, cramps, bloating and diarrhea were noted. 50% of subjects were classified as LNP and intolerant, 8% were tolerant LNP, 15% were digesters reporting symptoms and 27% were asymptomatic digesters. The 45 LNP subjects that reported symptoms of intolerance were fed 275 g (315 mL) of lactose-containing milk (L+) or lactose-free (L–) milk on 3 different days. Fasted, subjects were fed either L+ milk twice and L– milk once, or L– milk twice and L+ milk once, in a randomized design and symptoms were recorded. Sixty-seven percent of subjects reported symptoms from L+ milk and not L– milk. Thirty-three percent reported symptoms with both types of milks. These results suggest that although approximately 1/3 of African Americans claim intolerance or a symptomatic response after ingestion of milk, the lactose content cannot be the sole cause of the symptoms. The authors suggest that it was more likely that varying degrees of intestinal adaptation explain the symptoms experienced by this population. 

In 1994, Rao et al. reported symptoms of lactose intolerance in 98 adults, 20–89 years of age [2]. Fifty-two subjects were black, 46 were white, 48 were male, and 50 were female. Each subject was fed 16.5 g lactose in 360 mL of milk. 1 h after milk consumption, each subject ate a standardized breakfast (two hard-boiled eggs, one slice of toast, one fresh apple, one fresh banana, 250 mL of apple juice, once cup of black coffee, two sugar packets, and one pat of margarine). Breath hydrogen was monitored for 5 h and those excreting greater than 20 ppm of breath hydrogen at any time were classified as LNP. Forty-six percent of adults over 50 years were LNP; while only 26% of younger adults were LNP. There were 2.4 times more black LNPs under 50 years and 3.6 times more black LNPs in adults over 50 years. Of the LNPs, 63% reported symptoms and 3% reported severe symptoms. The authors did not report correlations of symptoms with age, race or gender. They did correlate abdominal cramps and flatulence with breath hydrogen, which did increase with age. The study leaves unanswered why breath hydrogen increased with age, but does not provide new information on LI related to gender and race.

Also in 1994, Rosado et al. reported on lactose maldigestion and milk intolerance in rural and urban Mexico [3]. All 926 subjects received two treatments: 240 mL of whole milk for children and 360 mL of whole milk for adults, and the same amounts of 90% lactose-hydrolyzed milk. Seventy-two percent of the subjects received a water solution with 8 g of lactulose for children and 10 g of lactulose for adults, as a positive control. Lactose maldigestion was detected by hydrogen breath test. Symptoms including headache, gas/flatulence, abdominal cramps, leg pain, and diarrhea were rated on a questionnaire for four hours after the administration of the test. LNP was higher in subjects from central and southern Mexico than Northern Mexico. With regard to age, 5% <4, 18% 4–8, 22% 8–13, 30% of adults were LNP. Major symptoms were present in 0–11% of children older than 4 years of age, and 7–17% of subjects 13–60 years of age. There was no significant difference in LNP or milk intolerance between rural and urban populations. The authors conclude that milk intolerance due to lactose maldigestion in Mexico is not as severe as it has been previously suggested. It does not appear in this Hispanic/Mexican sample that lactose intolerance is more severe than in other LNP populations. 

In 1994, Suarez and Savaiano assessed lactose digestion and tolerance in 20 adults (ages 20–40 years), and 20 elderly (65 years or older) Asian-Americans [15]. Subjects were evaluated for baseline lactose consumption, fecal beta-galactosidase activity and LNP, and then fed 0.5 g of lactose per kilogram of body weight in a 400 mL aqueous solution after a 12 h fast. Symptoms of headache, flatulence, stomach pain/cramps, and diarrhea/loose stool were rated on a 0 (none)–5 (severe) scale each hour for 8 h after the test drink was administered. No significant differences in total hydrogen production, flatulence or fecal beta-galactosidase activity were found between the adult and elderly groups. The mean of the summation of the symptom scores over 8 h for the adult group were: headache, 0.0; flatulence, 7.8; stomach pain/cramps, 2.6; and diarrhea/loose stool, 2.2. For the elderly group, the mean summation of the symptoms scores were: headache, 0.8; flatulence, 9.0; stomach pain/cramps, 1.3; and diarrhea/loose stool, 2.8. Thus, the results suggest that tolerance does not decrease with age. These results are novel because the dose did account for differences in body size, which sets this study apart from others published prior to this date. The results suggest that when the dose of lactose is proportional to body size, there is no relationship between age and LI. The study is limited by a small population, one racial group and one dose of lactose.

Gremse et al. reported a study of LNP and symptoms in 30 children, ages 3 to 17 [4]. Eleven were male, 19 were female, 16 were African American and 14 were Caucasian. All 30 subjects were fed 240 mL of lactose-free and lactose-containing milk daily for 14 days each in a double-blinded cross-over design. Symptoms were recorded in diaries. Twelve grams of lactose or more caused significantly increased abdominal pain in children with LNP. Twenty-three subjects reported more abdominal pain after ingesting the lactose-containing milk, while seven subjects reported less pain or no change in symptoms. The dose of 240 mL of milk caused symptoms in most, but not all, of the subjects in this study. The authors also conclude that increased abdominal pain is associated with the ingestion of 12 g of lactose daily in children with lactose maldigestion. No comparisons to adult populations were made. While children in this study experience significant LI symptoms, the fed dose did not account for body weight, which is especially important when the sample includes very small children. 

Asmawi et al. assessed LNP and lactose intolerance in 300 Malaysians, ages 18–49 years [6]. One hundred of the subjects were native Malaysia, 100 were Chinese immigrants and 100 were Indian immigrants. Urinary galactose and symptoms of LI (bloating, flatulence, borborygmi, abdominal pain, nausea, vomiting, diarrhea and loose stools) were evaluated following a 50 g lactose dose. Eighty-eight percent of the Malays, 91% of the Chinese, and 83% of the Indians were LNP, and 77% of the Malays, 71% of the Chinese, and 62% of Indians experienced symptoms. Flatulence and borborygmi were the most common symptoms in all three ethnic groups. The 50 g dose of lactose is that found in a quart of milk and thus represents an un-physiological load. It is not surprising that the majority of subjects experienced symptoms. It is perhaps more remarkable that 23% of the Malaysians, 29% of the Chinese immigrants and 38% of the Indian immigrants did not experience symptoms after drinking a quart of milk. This result emphasizes that LI symptoms cannot be linked to these racial demographics, even after the consumption of a large unphysiological dose of lactose.

In 2009, Laaksonen et al. reported on LNP (C/C genotype), consumption of milk products and intakes of milk nutrients in 3596 Finnish children, ages 3, 6, 9, 12, 15 and 18 years [16]. Dietary data came from the Cardiovascular Risk in Young Finns Study, with the first cross-sectional survey in 1980 and subsequent surveys in 1983, 1986, 1989, 1992 and 2001 using the same dietary questionnaire. There were lower daily calcium and lactose intakes observed for C/C genotype females older than 9 years. The C/C genotype was associated with lower daily calcium and lactose intake in both sexes in adulthood. Intakes of calcium, lactose, protein and vitamin D from milk were significantly lowest for the C/C genotype in both sexes, however, the intake of calcium, lactose, protein and vitamin D from cheese or sour milk products did not differ between the genotypes. Correspondingly, lactose content of milk and milk products consumed was lowest for C/C genotype individuals, but only statistically significant for females. Inadequate calcium intake was greatest in C/C genotype for females, significant at 9 and 15 years, and adulthood, and for males, significant in adulthood. Finally, over half of females and 1/3 of males with C/C genotype reported not drinking milk in adulthood and those with C/C genotype reported following low-lactose or milk-free diet more often than other genotypes, C/T or T/T. While this study did not measure symptoms, it does demonstrate that LNP is correlated with milk avoidance. Interestingly, the consumption of sour milk products did not differ between the lactase genotypes in either sex or at any age. Since consumption of sour milk products did not differ between lactase genotypes, maldigesters were not limiting consumption of these products. Therefore, this study suggests that the use of cultured or fermented dairy foods may be a strategy for avoiding lactose intolerance symptoms. 

Nicklas et al., (2011) reported self-perceived LI and its relationship to calcium, dairy food intake, hypertension and diabetes in 3452 adults, ages 19 to 70 years [7]. Twelve point three percent of respondents perceived themselves to be LI. Of these, 7.8% were non-Hispanic whites, 20.1% were non-Hispanic blacks, 8.8% were Hispanics, 76.13% were female, and 23.87% were male. The self-perceived lactose-intolerant respondents reported significantly lower calcium intake from dairy foods, and significantly higher rates of physician-diagnosed diabetes and hypertension. While Nicklas et al did not report symptoms, the avoidance of milk could be related to real or perceived intolerance, or other factors including cultural norms, and food economics. 

In 2011, Tomba et al. reported on whether subjective perception of LI is correlated with psychological profile [17]. One-hundred and two consecutive patients were fed a 15 g lactose challenge. Breath hydrogen determined LNP. Symptom response to the challenge, and a psychological profile was reported. LNP was present in 18% of patients and LI symptoms were reported by 29% of patients. The severity of symptoms was higher in patients with an altered level of somatization. The authors suggest that a psychological component may be relevant to an individual’s perception of LI, which would result in a reduced dairy food intake. Thus, certain individuals may avoid lactose for reasons other than LI. It is possible that psychological profile could vary by age, gender or race, leading to differences in perceived LI. This is an area for future research.

In 2013, Savaiano et al. reported a randomized, double-blind, parallel group, placebo-controlled study that assessed whether a galacto-oligosaccharide, RP-G28, can improve digestion and reduce symptoms of LI [18]. Eighty-five subjects with LI (as demonstrated by an inclusion test with milk) were 18–64 years old. Forty-two percent were male, 38% were Asian, 26% were African-American, 15% were white, and 21% were another race. Fifty-seven patients received the RP-G28 treatment and 28 received a placebo for 35 days while avoiding dairy products. Subjects reintroduced dairy into daily diets post-treatment and were followed for 30 additional days. LNP was measured by breath hydrogen and symptom improvements were measured by patient self-report. At the baseline, day 36 (end of the treatment) and day 66 (30 days post treatment), a 25 g in-clinic lactose dose was administered, and symptoms of abdominal pain, bloating, flatulence, diarrhea, and cramping were. Lactose digestion, as measured by breath hydrogen, and corresponding symptoms improved following RP-G28 treatment, both at the end of treatment and 30 days post-treatment. Fifty percent of RP-G28-fed subjects with abdominal pain at baseline reported no abdominal pain at the end of treatment and 30 days post-treatment. RP-G28-treated subjects were also 6 times more likely claim lactose tolerance post-treatment once dairy foods had been re-introduced to their diets. No differences in improvement of LI based on demographics were reported. Almost all subjects showed improvement, suggesting that solutions for LI individuals do not vary by gender, age or race. 

In 2014, Baadkar, Mukherjee and Lele reported the influence of age, gender and genetic variants on LNP, lactose intolerance and milk intake in 205 healthy adult Asian Indians, 18–73 years. [5]. One hundred and twenty-three subjects were male and 82 were female. Eight percent of subjects were Northern Indians, 7% were Southern Indians and 85% were Western Indians. Subjects ingested 25 g of lactose in water and blood sugar was measured 30 min after lactose intake. Bloating, flatulence, borborygmi, abdominal pain, nausea, vomiting and diarrhea were recorded. Forty-two point nine percent of subjects over 50-year-old subjects developed symptoms as compared to only 29.7% of younger subjects. LNP was similar between the two age groups at 84.4% and 83.1%, respectively. Further, 45.1% of the female subjects, but only 27.6% of the male subjects developed symptoms. Females and subjects with the LNP genotype (-13910CC) had significantly higher incidence of LI. Finally, the quantity and frequency of historical milk intake was lower for subjects with the LNP genotype. These results suggest an age and gender difference in symptom response to the same lactose dose. However, the use of a single dose of lactose regardless of body weight limits the interpretation of the data. Body size should be accounted for in the dose, as older individuals and females may be smaller than younger males.

## 3. Conclusions

We conclude that there is insufficient evidence to support any unique associations between LI symptoms and gender, age or race, except for the expected differences due to genetically determined LNP. Research to date has not assessed gender and racial differences in LI in studies where lactose was administered based on body weight. Only one study assessed LI among young and elderly LNP, feeding lactose based on body weight and showed no difference [15]. The suggestive and interesting findings of Baadkar [5] need replication in additional populations, with lactose or milk dose fed based on body weight. Further, it would be interesting to correlate LI symptoms with prior milk consumption. Prior milk consumption is a likely indicator of colonic adaptation, and a probable confounder in the relationship between LNP and symptom response. 

## Figures and Tables

**Table 1 nutrients-10-01956-t001:** Publications reporting on lactose dose, gender, age, race/ethnicity and symptom-response.

Author	Lactose Dose	#Subjects	Gender	Age (Years)	Race/Ethnicity	Maldigesters	Symptoms Present
Stephenson & Latham 1974 [9]	50 g in 200 mL water	35	19 femals 16 males	25 median	25 Caucasians 10 non-Caucasians	45%	93.75% maldigesters 80% non-caucasians maldigesters Severity varied Diarrhea, gas, bloating, cramps Mild, moderate, severe for 8 hours post dose Most maldigesters consumed 15–30 g lactose without severe symptoms
Rorick & Scrimshaw 1979 [10]	11 g in 240 mL drink vs. 0 g	87	67 femals 20 males	60–97	N/A	26% Highest in Black, Jewish & Italian subjects Lowest in northern & western Europeans	54 symptoms reported by 25 subjects 39 mild, 11 moderate, 4 severe
Haverberg, Kwon & Scrimshaw 1980 [11]	50 g in 250 mL water 11 g in 240 mL milk 22 g in 480 mL milk 0 g	110	N/A	14–19	58 Blacks 44 Caucasians 8 Latin-Americans	61% 83% Black 62% Hispanic 32% Caucasian 61% adolescents	39% maldigesters after 480 mL w/22g lactose 22% maldigesters after 480 mL lactose-free 19% digesters after 480 mL w/22g lactose 32% digesters after 480 mL lactose-free
Kwon, Rorick & Scrimshaw 1980 [12]	50 g in 250 mL water 11 g in 240 mL milk 22 g in 480 mL milk 0 g	87	N/A	14–19	26 blacks 56 caucasian 5 asian-American	52% 81% Black 80% Asian-American 36% Caucasian 52% adolescents	No maldigesters had symptoms after 240 mL 16% maldigesters after 480 mL milk
Rosado, Allen & Solomons 1987 [13]	18 g in 360 mL whole milk	25	5 females 20 males	19–48	2 Blacks 13 Caucasians from n. Europe 9 other Caucasians	36%	Same response in digesters & maldigesters
Ito & Kimura 1993 [14]	15 g in 115 mL water 1x/day for 6 days	24	Male	25–45	Japanese	92%	29% of all subjects 23% of maldigestors Discomfort & diarrhea
Johnson et al. 1993 [1]	25 g in 200–300 mL water; then 275 g (315 mL) lactose-containing or lactose digested milk on different days	164	N/A	12–40	N/A	50% LI maldigesters 8% tolerant maldigesters	33% with both milk types 15% LI digesters 27% asymptomatic digesters
Rao, Bello & Brown 1994 [2]	16 g in 360 mL milk	98	50 female 48 male	20–89	52 Blacks 46 Caucasians	2.4x more Black maldigesters under 50 years 3.6x more Black maldigesters in adults over 50 years	63% maldigesters 3% of total
Rosado et al. 1994 [3]	240 mL (children) & 360 mL (adults) whole milk vs. lactose-hydrolyzed milk	926	N/A	Children & adults	N/A	N/A	0–11% of children > 4 years 7–17% of 13–60 year old subjects No symptoms from lactose-hydrolyzed milk
Suarez & Savaiano 1994 [15]	0.5 g per kg body weight	40	N/A	20–40 vs. >65	N/A	N/A	Highest symptom score for flatulence Headache, flatulence, stomach pain/cramps, diarrhea/loose stool No difference between adults and elderly
Gremse et al. 2003 [4]	240 mL lactose-hydrolyzed or lactose-containing milk	30	19 female 11 male	3–17	16 Blacks 14 Caucasians	100%	23/30 subjects reported increased abdominal pain w/ lactose-containing milk
Asmawi et al. 2006 [6]	50 g	300	N/A	18–49	88% Malays 91% Chinese 83% Indians	N/A	77% Malays, 71% Chinese, 62% Indians Flatulence and borborygmi most common Bloating, flatulence, borborygmi, abdominal pain, nausea, vomiting, diarrhea, loose stools
Laaksonen et al. 2009 [16]	Milk drinking habits and genetics	3596	N/A	3–18	N/A	N/A	Over 1/2 females and 1/3 males with C/C genotype reported not drinking milk in adulthood; those with C/C genotype reported following low-lactose or milk free diet more often than other genotypes
Nicklas et al. 2011 [7]	LI perception	3452	N/A	19–70	N/A	N/A	Perceived LI: 76.13% female 23.87% male 7.8% non-hispanic Whites 20.1% non-hispanic Blacks
Tomba et al. 2012 [17]	15 g	102	N/A	N/A	N/A	18%	29% total Symptoms unrelated to maldigestion Severity of symptoms ↑ in patients with altered level of somatization Abdominal pain, borborygmi, bloating, flatulence, nausea
Savaiano et al. 2013 [18]	15–25 g Galacto-oligosaccharide	85	49 female 36 male	18–64	22 Blacks 13 Whites 32 Asians 18 other	100%	LI verified by milk challenge Abdominal pain, bloating, flatulence, diarrhea & cramping Reduced abdominal pain & improved tolerance
Baadkar, Mukherjee & Lele 2014 [5]	25 g	205	82 female 123 male	18–78	16 northern Indians 14 southern Indians 175 western Indians	82.9% in females 84.6% in males	LI 45.1% in females LI 27.6% in males 42.9% subjects over 50 years 29.7% subjects under 50 years Bloating, flatulence, borborygmi, abdominal pain, nausea, vomiting & diarrhea

N/A is no data provided. LI is lactose intolerance.

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
