# Peer review of "Gender, Age, Race and Lactose Intolerance: Is There Evidence to Support a Differential Symptom Response? A Scoping Review"

_nutrients, 2018, doi:10.3390/nu10121956_

Round 1
Reviewer 1 Report
We used to call studies like this a "mega analysis". It tries to make sense out of a wide array of data. Thank you!
Author Response
We thank the reviewer for their review.
Reviewer 2 Report
In Line 187: Lactose-hydrolyzed milk: Does this amount to a lactose free control? How many subjects were given lactose hydrolyzed and lactose containing milk needs to be indicated.
In the Laaksonen study, In line 218there is a mention of lower daily calcium and lactose intake for C/C genotype subjects. What is the basis of this lowered intake, is it a self driven dose or doctor's recommendation. Or is it based on historical milk consumption records, how were they maintained.
How is the conclusion drawn in Line 223 and 224
In the 2013 Saviaono study, why isn;t the Hydrogen breath test used as an indicator 30 days post treatment. If yes, a mention would serve as a postive reinforcement to the self assessment of symptoms by the subjects.
Why is the paragraph with Baadkar study in a smaller font?
Line 260: 18-173 years?? Do you mean 18-73 years
Author Response
In Line 187: Lactose-hydrolyzed milk: Does this amount to a lactose free control? How many subjects were given lactose hydrolyzed and lactose containing milk needs to be indicated.
Yes. The authors state that the milk is lactose-free. We have edited the manuscript to ‘lactose-free milk’. The study was a randomized cross-over with all 30 subjects consuming both milks for 14 days. We have added text to better describe this protocol.
In the Laaksonen study, In line 218there is a mention of lower daily calcium and lactose intake for C/C genotype subjects. What is the basis of this lowered intake, is it a self driven dose or doctor's recommendation. Or is it based on historical milk consumption records, how were they maintained.
This dietary data comes from the Cardiovascular Risk in Young Finns Study with the first cross-sectional survey in 1980 and subsequent surveys in 1983, 1986, 1989, 1992 and 2001 using the same dietary questionnaire. It is self-driven.
How is the conclusion drawn in Line 223 and 224
Since consumption of sour milk products did not differ between lactase genotypes, maldigesters were not limiting consumption of these products. Hence, a recommendation could be made to promote these products to improve diet quality (calcium, protein and vitamin D). We have added a sentence to the manuscript to further explain this idea.
In the 2013 Saviaono study, why isn;t the Hydrogen breath test used as an indicator 30 days post treatment. If yes, a mention would serve as a postive reinforcement to the self assessment of symptoms by the subjects.
Results from a hydrogen breath test were reported at Day 0 and Day 36, but not at day 66. Day 66 hydrogen breath testing was not conducted.
Why is the paragraph with Baadkar study in a smaller font?
We have edited.
Line 260: 18-173 years?? Do you mean 18-73 years
Yes. Corrected.
Reviewer 3 Report
This manuscript is a scoping review on the topic of lactose intolerance. The topic of lactose intolerance is important in the field of nutritional health, given more than 50% of the world’s adult population are lactose intolerance at some point in their lives.
HOWEVER, THE SPECIFIC PURPOSE OF THIS REVIEW WORK IN TERMS OF HOW IT HELPS ADVANCE THE KNOWLEDGE BASE IN THE FIELD OF LACTOSE INTOLERANCE IS UNCLEAR.
A SECOND MAJOR DRAWBACK OF THIS REVIEW WORK IS THAT IT DOES NOT DISCUSS THE NEWER RESEARCH AND DIRECTION IN THE FIELD THAT WERE PUBLISHED IN LAST FOUR YEARS! MUCH HAS BEEN REPORTED ON THE GENOTYPES ASSOCIATED WITH LACTOSE INTOLERANCE, ROLE OF EPIGENETICS, AND THE GUT MICROBIOTA IN LACTOSE INTOLERANCE AND BEYOND. NONE OF THESE NEWER ASPECTS WERE CAPTURED IN THE MANUSCRIPT.
MANY OLD NUTRITIONAL HEALTH RESEARCH STUDIES ARE CONTRADICTED BY NEWER RESEARCH STUDIES AND CONCEPTS, GIVEN THAT THE NUTRITION SCIENCE FIELD HAS REVOLUTIONIZED IN THE POST-GENOMIC ERA. A PUBMED SEARCH OF REVIEW PAPERS PUBLISHED ON THE TOPIC OF LACTOSE INTOLERANCE YIELDED 605 ARTICLES ON NOVEMBER 8 OF 2018. OF THESE 605, 82 ARE PUBLISHED DURING OR AFTER 2014.
SINCE THE AUTHORS ONLY REVIEWED RESEARCH PUBLISHED UNTIL 2014 AND DID NOT DISCUSS ANYTHING ABOUT THE COLLECTIVE INFORMATION GENERATED BY THESE 82 REVIEWS (several of these published in the journal Nutrients) OR BY LANDMARK RESEARCH ARTICLES PUBLISHED BEFORE 2014, IT IS HIGHLY LIKELY THAT THIS REVIEW WORK DOES NOT ADD ANY NEW INFORMATION OR PERSPECTIVE THAT CAN ADVANCE THE CURRENT UNDERSTANDING IN THE FIELD.
The above statements of the reviewer are further supported by the following points:
1) The conclusion is brief and states that the 14 studies reviewed did not conclusively associate Lactose Intolerance symptoms with gender, age, or race. Generally speaking, association studies are not meant to support any unique relationship between a clinical symptoms and demographic characteristics, but rather indicate a possibility that must be verified by mechanistic research. Association studies generate new hypothesis to be pursued by future research and many of the research discussed in this work effectively does that.
2) Although some of the studies reviewed here indicated a possible relationship between lactose intolerance and gender, age or race, and based on these studies some may currently perceive that such possibility likely exist, these perceptions were not even mentioned as an issue in the concluding statements of the NIH Consensus Conference on Lactose Intolerance and Health, held in 2010. This point further highlights why this work may not advance the field in any way.
3) The current standing that emerged from the newer review works and NIH conference may be summarized as follows, and in the light of which the scope of this review manuscript appears to be somewhat irrelevant.
One of the current standing in the field is that differences in lactase activity are due to a genetic polymorphism. There are three genotypes: homozygous lactase persistent (LP), homozygous lactase non-persistent (LNP), and heterozygotes. Intermediate enzyme activity in the heterozygotes implied that cis-acting allelic variations as a regulatory agent neighboring the lactase gene are responsible for the polymorphism. While much of the recent genetic and anthropologic literature on lactose intolerance is focused on LP, all the phenotypes appear to be under genetic control with lactase persistence being the mutation that is seemingly the result of generations of continued milk consumptions as an historic example of positive selection. THE CURRENT NEED IN THE FIELD IS TO DIRECT educational efforts at encouraging individuals with lactase non-persistence to consume adequate amounts of calcium and Vitamin D either as forms of dairy foods with lowered lactose content, such as hard cheese, yogurt, and lactose hydrolyzed products, modest amounts of milk with meals which dilute the lactose and delay gastric emptying or as supplements of calcium and vitamin D.
Author Response
This manuscript is a scoping review on the topic of lactose intolerance. The topic of lactose intolerance is important in the field of nutritional health, given more than 50% of the world’s adult population are lactose intolerance at some point in their lives. HOWEVER, THE SPECIFIC PURPOSE OF THIS REVIEW WORK IN TERMS OF HOW IT HELPS ADVANCE THE KNOWLEDGE BASE IN THE FIELD OF LACTOSE INTOLERANCE IS UNCLEAR.
The purpose of this scoping review is to dispel a commonly held myth that women, the elderly and certain racial groups have a greater incidence of LI, independent of LNP status. We have edited to attempt to more clearly state this purpose.
A SECOND MAJOR DRAWBACK OF THIS REVIEW WORK IS THAT IT DOES NOT DISCUSS THE NEWER RESEARCH AND DIRECTION IN THE FIELD THAT WERE PUBLISHED IN LAST FOUR YEARS! MUCH HAS BEEN REPORTED ON THE GENOTYPES ASSOCIATED WITH LACTOSE INTOLERANCE, ROLE OF EPIGENETICS, AND THE GUT MICROBIOTA IN LACTOSE INTOLERANCE AND BEYOND. NONE OF THESE NEWER ASPECTS WERE CAPTURED IN THE MANUSCRIPT.
Unfortunately, none of this work is directed specifically at age, gender and racial differences in LI independent of LNP status.
MANY OLD NUTRITIONAL HEALTH RESEARCH STUDIES ARE CONTRADICTED BY NEWER RESEARCH STUDIES AND CONCEPTS, GIVEN THAT THE NUTRITION SCIENCE FIELD HAS REVOLUTIONIZED IN THE POST-GENOMIC ERA. A PUBMED SEARCH OF REVIEW PAPERS PUBLISHED ON THE TOPIC OF LACTOSE INTOLERANCE YIELDED 605 ARTICLES ON NOVEMBER 8 OF 2018. OF THESE 605, 82 ARE PUBLISHED DURING OR AFTER 2014.
Actually, we disagree. We find great congruence between older work and newer publications. This congruence is sometimes difficult to ascertain based on the historical 50g dose used, and the difficulty with double-blinding in many older studies.
SINCE THE AUTHORS ONLY REVIEWED RESEARCH PUBLISHED UNTIL 2014 AND DID NOT DISCUSS ANYTHING ABOUT THE COLLECTIVE INFORMATION GENERATED BY THESE 82 REVIEWS (several of these published in the journal Nutrients) OR BY LANDMARK RESEARCH ARTICLES PUBLISHED BEFORE 2014, IT IS HIGHLY LIKELY THAT THIS REVIEW WORK DOES NOT ADD ANY NEW INFORMATION OR PERSPECTIVE THAT CAN ADVANCE THE CURRENT UNDERSTANDING IN THE FIELD.
We have completed a comprehensive review based on the key words as described in 20-36 of the paper. We would very much appreciate studies that combine LI symptom evaluation, LNP status with gender, age and race. Unfortunately, they did not appear in our comprehensive search strategy. Scoping reviews often find areas for future research. We appear to have found one.
The above statements of the reviewer are further supported by the following points:
1) The conclusion is brief and states that the 14 studies reviewed did not conclusively associate Lactose Intolerance symptoms with gender, age, or race. Generally speaking, association studies are not meant to support any unique relationship between a clinical symptoms and demographic characteristics, but rather indicate a possibility that must be verified by mechanistic research. Association studies generate new hypothesis to be pursued by future research and many of the research discussed in this work effectively does that.
2) Although some of the studies reviewed here indicated a possible relationship between lactose intolerance and gender, age or race, and based on these studies some may currently perceive that such possibility likely exist, these perceptions were not even mentioned as an issue in the concluding statements of the NIH Consensus Conference on Lactose Intolerance and Health, held in 2010. This point further highlights why this work may not advance the field in any way.
3) The current standing that emerged from the newer review works and NIH conference may be summarized as follows, and in the light of which the scope of this review manuscript appears to be somewhat irrelevant.
One of the current standing in the field is that differences in lactase activity are due to a genetic polymorphism. There are three genotypes: homozygous lactase persistent (LP), homozygous lactase non-persistent (LNP), and heterozygotes. Intermediate enzyme activity in the heterozygotes implied that cis-acting allelic variations as a regulatory agent neighboring the lactase gene are responsible for the polymorphism. While much of the recent genetic and anthropologic literature on lactose intolerance is focused on LP, all the phenotypes appear to be under genetic control with lactase persistence being the mutation that is seemingly the result of generations of continued milk consumptions as an historic example of positive selection. THE CURRENT NEED IN THE FIELD IS TO DIRECT educational efforts at encouraging individuals with lactase non-persistence to consume adequate amounts of calcium and Vitamin D either as forms of dairy foods with lowered lactose content, such as hard cheese, yogurt, and lactose hydrolyzed products, modest amounts of milk with meals which dilute the lactose and delay gastric emptying or as supplements of calcium and vitamin D.
We thank the reviewer for these comments. We most strongly agree with the reviewer for the need to educate individuals to consume dairy foods while avoiding symptoms of lactose intolerance. Education is a primary purpose of this review. Misconceptions about age, gender and race related to LI are a key barrier to effective education. We have edited the manuscript with the aim of more clearly describing why this scoping review is relevant at the present time. As the reviewer points out, variations in symptoms of lactose intolerance result not simply from the three genotypes, but vary with diet, transit and the microbiome. Studies indicate that dose, transit, meal-feeding, microbiome adaptation and other less well described factors such as genetic variants of beta-casein influence symptom response. Symptoms may also be partially determined by the residual lactase which has a normal distribution in the three phenotypes. This was not a scoping review to summarize the important genetic information that has been published that has significantly enhanced our understanding of the molecular biology behind LP and LNP. Rather, it has a more practical purpose, to advise dietitians, physicians and other nutrition educators that the available information does not support a differential symptom-response based on gender, age or race (except for LNP status). Rather, dose, diet (including dietary adaptation of the microbiome), body size and LNP are the likely primary drivers of intolerance symptoms.
Round 2
Reviewer 3 Report
The authors have partially responded to the previous concerns and have now mentioned the NIH Consensus Conference on Lactose Intolerance (LI) that was held in 2010.
However, the primary concern of the reviewer stated previously “THE SPECIFIC PURPOSE OF THIS REVIEW WORK IN TERMS OF HOW IT HELPS ADVANCE THE KNOWLEDGE BASE IN THE FIELD OF LACTOSE INTOLERANCE IS UNCLEAR” has not been addressed.
The authors argued that “The purpose of this scoping review is to dispel a commonly held myth that women, the elderly and certain racial groups have a greater incidence of LI, independent of LNP status. We have edited to attempt to more clearly state this purpose.” They also now included the following statement in the paper:- “Misconceptions about age, gender and race related to LI are a barrier to effective education and dietary selection.”
But they have not cited any evidence to support that misconceptions about age, gender and race related to LI actually is prevalent in clinical practices in the United States or another country. The authors did not establish why they think the misconception is prevalent and that it indeed creates a barrier to effective education in LI individuals. SINCE THE ENTIRE PAPER IS DIRECTED TO ADDRESS THIS PERCEIVED PROBLEM, AUTHORS MUST PROVIDE STRONG EVIDENCE TO FIRST AUTHENTICALLY LAY OUT THE PROBLEM AND THEN ADDRESS IT THROUGH THEIR SCOPING REVIEW. Just because some older research papers reported the possibility of (all of the cited studies are association studies) age, gender and race related differential LI prevalence does not mean that this perception has been adopted in a widespread manner and being currently practiced causing any practical challenges. There has been no follow-up mechanistic studies to validate those association studies that is available in the literature. Moreover, there are many newer papers that have in fact shown LI is related to genotype differences. Therefore, in absence of specific evidence, it remains unclear why the authors believe that this perception is currently prevalent and continues to create a barrier to education and diet selection related to LI. TO SUMMARIZE, IN ABSENCE OF THIS EVIDENCE, THE PURPOSE OF THE REVIEW REMAINS UNCLEAR AND HENCE, IT DOES NOT ADVANCE THE KNOWLEDGE BASE IN THE FIELD IN ANY WAY.
Some suggestions on what type of evidence may help establish the problem:
Are there multiple citations showing this perception does exist at present times and continues to be a problem in clinical nutrition and medical practices.

Did the NIH consensus conference (2010 or a more recent one) include or discuss this perception as being a potential problem in their conclusion/summary statements/reports? NIH typically reviews current status in the field from time to time by gathering a broad range of experts (both researchers and practitioners) and if they did not discuss this as a problem, perhaps the experts did not think that this perception exists and is a potential problem?
Is there any official guideline (dietary or medical) related to this perception that are being followed currently in clinical practices?
Author Response
We again thank the reviewer for their comments. Hopefully, we have further clarified our purpose for writing this review, and we have more clearly communicated that purpose in the introduction. There are several research reports indicating that women, the elderly and certain racial groups perceive themselves to be more lactose intolerant than would be expected based on LNP. Further, evidence is quite clear that perception influences dietary choice. Thus, perceived lactose intolerance appears to be the actual driver of reduced calcium intakes. These perceptions are given further support by studies that feed a single dose of lactose, not accounting for differences in body weight between men and women in particular. Hence, the introduction has been rewritten to also focus on perceived LI as there is considerable body of literature indicating the relationship between perception and diet choice.